# Outbreak analysis with a logistic growth model shows COVID-19 suppression dynamics in China

Yi Zou[1], Stephen Pan[1], Peng Zhao[1], Lei Han[1], Xiaoxiang Wang[2], Lia Hemerik[3], Johannes Knops[1], Wopke van der Werf[4]*

1 Department of Health and Environmental Sciences, Xi'an Jiaotong-Liverpool University, Suzhou, China, 2 School of the Environmental Science and Engineering, Southern University of Science and Technology, Shenzhen, China, 3 Wageningen University, Biometris, Wageningen, The Netherlands, 4 Centre for Crop Systems Analysis, Wageningen University, Wageningen, The Netherlands

* wopke.vanderwerf@wur.nl

## Abstract

China reported a major outbreak of a novel coronavirus, SARS-CoV2, from mid-January till mid-March 2020. We review the epidemic virus growth and decline curves in China using a phenomenological logistic growth model to summarize the outbreak dynamics using three parameters that characterize the epidemic's timing, rate and peak. During the initial phase, the number of virus cases doubled every 2.7 days (range 2.2–4.4 across provinces). The rate of increase in the number of reported cases peaked approximately 10 days after suppression measures were started on 23–25 January 2020. The peak in the number of reported sick cases occurred on average 18 days after the start of suppression measures. From the time of starting measures till the peak, the number of cases increased by a factor 39 in the province Hubei, and by a factor 9.5 for all of China (range: 6.2–20.4 in the other provinces). Complete suppression took up to 2 months (range: 23-57d.), during which period severe restrictions, social distancing measures, testing and isolation of cases were in place. The suppression of the disease in China has been successful, demonstrating that suppression is a viable strategy to contain SARS-CoV2.

## Introduction

The coronavirus SARS-CoV2 emerged in Wuhan, Hubei province, China, in late 2019. From there it spread, first in Hubei, then across China during the spring holiday, and finally across the world. Currently (as of 8 April 2020), the virus has been reported from 212 countries areas or territories and the cumulative number of cases outside China has exceeded 1.2 million [1]. In mainland China, few new cases have been reported since 18 March, and most new cases have originated from outside China [2]. Thus, the SARS-CoV-2 outbreak in China appears to be, for now, under control.

Virtually the entire global population is susceptible to SARS-CoV2 infection and no vaccine is available yet. Unless a successful NPI (Non-Pharmaceutical Intervention) strategy is

**Data Availability Statement:** Data used in this study are publicly accessible from the R package nCov2019 (https://github.com/GuangchuangYu/nCov2019).

**Funding:** The authors received no specific funding for this work.

**Competing interests:** The authors have declared that no competing interests exist.

implemented in the early stages of transmission, an exponential global proliferation threatens to overwhelm the health care systems of many countries. Three NPI strategies are being discussed for managing the current COVID-19 epidemic: (1) suppression, (2) mitigation, and (3) containment [3]. In **suppression**, strict measures are taken to reverse the epidemic spread, essentially by bringing the effective reproduction number $R_e$ (the number of new cases per existing case) below one [4, 5]. Social distancing is a key factor in suppression [6]. In **mitigation** the aim is not to necessarily stop all transmission, but rather to reduce the rate of transmission and in effect lower the number of infected people at any given time [3, 7]. It has been suggested that mitigation strategies might prevent inundation of the health care system by "flattening" the peak of sick people" [3]. However, even in the most optimistic scenarios for mitigation, healthcare capacity is likely to be still seriously overwhelmed, as it was in Wuhan in February 2020 and in Italy in March 2020. Herd immunity has been suggested as a component of mitigation, but is only a viable option once a vaccine is available because a sizeable proportion of infected people develops serious symptoms and needs hospitalization or intensive care [8]. The proportion of **infected** people requiring hospital treatment is in the order of several percent but not well known because the reporting fraction of infected cases is not well known, and it also depends on age [8]. The proportion of people with confirmed infection needing hospitalization or intensive care varies between countries according to the definition of "confirmed infection", the criteria used for admission to hospitals, the age distribution, and other factors. This proportion was initially estimated at 19% in China, based on data of the Chinese Center for Disease Control and Prevention [8]. The proportion of all infected people requiring hospitalization (with infection reported or unreported) is lower than this 19%, but is uncertain due to inaccurate knowledge of the proportion of infected people being reported as infected. The proportion of unreported cases was estimated at 86% before measures and at 16–21% after measures in the whole of China by Li, Pei [9]; and an estimate of 59% or more was estimated for Wuhan by Wang, Liu [10]). If underreporting is accounted for, the proportion of infected people (reported + unreported) requiring hospitalization is in the order of 3% or more. **Containment** is based on intensive surveillance of possible cases, testing, followed by isolation of infected people and their contacts [11]. Containment is only possible if the virus is not freely circulating in a population. Currently, because in many countries outside of China the virus is circulating within the population, these countries are practicing suppression. Policy makers may ask:

1. How long will it take for the epidemic to peak after suppression measures have been implemented?

2. What will the peak number of sickness cases be?

3. How long do suppression measures need to be maintained to suppress the virus to sufficiently low incidence to allow containment (search and quarantine flare-ups of the virus)?

Here we analyze with a phenomenological logistic model the epidemics of SARS-CoV2 in China and 20 of its provinces that reported more than 150 cases. The logistic model is widely used in ecology to analyze boom and bust population dynamics [12]. Logistic models are not as widely used in human disease epidemiology as more mechanistic compartment-based SIR and SEIR models [13–16] because the parameters lack a strict mechanistic interpretation in terms of transmission rate and disease etiology (e.g. latency period, incubation period and infectious period). However, this disadvantage is compensated by the usefulness of the parameters for providing a simple and intuitive description of the outbreak dynamics in time. We do not imply that logistic models can replace established epidemiological models, but we do argue that phenomenological models, given the urgency and severe consequences of the worldwide

SARS-CoV2 outbreak for public health management, have a place along mechanistic models to inform on disease dynamics [17] and support narratives on outbreak dynamics with simple metrics like relative rate of increase, doubling time, and time to the peak. Using data from China, logistic models shows the key disease dynamic parameters before and after suppression policies were implemented [18].

## Methods

Data were obtained from the open-source analytics tools, the R package "nCov2019" [19, 20], which retrieves data from the daily epidemic report in the National Health Commission of China [21]. We obtained time series data of total confirmed, total recovered, and total death cases for each provinces of China. We used the data starting on 21 January when reporting daily infected cases started at the national level, and up to 10 March 2020 when almost no new confirmed cases were reported in China [21]. We did not use data after 10 March to minimize the influence of cases introduced from outside China.

We used three-parameter logistic models to fit the time series of the total confirmed and of the total recovered cases. Parameters refer to the asymptotic value ($a$; number of cases), the inflection point of the curve ($b$; date) and a scale parameter ($c$; days).

$$T_t = \frac{a}{1 + \exp\left(\frac{b-t}{c}\right)}$$

$$C_t = \frac{a_2}{1 + \exp\left(\frac{b_2-t}{c_2}\right)}$$

where $T_t$ and $C_t$ are total infected and recovered cases at day $t$.

In addition, we assume a constant daily death rate $k$ (S1 Fig), which was calculated as the average of number of deceased each day, divided by the infected cases on that day:

$$k = \frac{1}{n}\sum_{i=1}^{n}\frac{D_i}{I_i}$$

Where $D_t$ and $I_t$ are the number of daily death cases and active infected cases at day $t$. We excluded data before 25 January and data where $I_t < 50$ in this calculation, as values are inaccurately estimated at an early stage of the outbreak (under-estimated denominator) and at low $I_t$ (high variability in the outcome). The model does not account for the delay between the moment of reporting disease and the moment of reporting death.

We then calculated the active infected cases, which can be expressed as

$$I_t = T_t - TD_t - C_t$$

Of which $TD_t$ is the number of total death at day t, which equals

$$TD_t = \sum_{i=1}^{t} k\ I_i$$

And therefore (see S1 Appendix for inference)

$$I_t = \frac{(T_t - T_{t-1}) - (C_t - C_{t-1}) + I_{t-1}}{1 + k}$$

where $T_t - T_{t-1}$ is the daily change in the number of infected cases, and $C_t - C_{t-1}$ is the daily change in the number of recovered cases.

Based on the fitted model, we then calculated the peak date of 1) active sick people, which is also the date of peak number of daily death as we assumed a fixed daily death rate in the model, 2) number of active sick cases during the peak, 3) the date of maximum increase in the number of infected cases and 4) the daily rate of increase on this date, 5) total infected cases on this date, 6) the relative rate on this date, 7) the end date of daily increase (<1) case (operationally the end of the epidemic), and 8) time from maximum increase till sick peak. Taking the date of level 1 public health emergency action as the implementation of suppression measures [22], which varied between 23 January to 25 January across provinces [23] (we set the median date, 24 January as the date for entire China), we then calculated 9) the delay from the action date until the sick peak, 10) the delay from the action date until the date at which the rate of increase peaked, 11) the time from suppression measure till the end date of daily increase in number of reported infections, 12) the ratio between sick cases at peak and total infected case at the action date, as well as 13) the same ratio considering a reporting delay of 6 days [9], i.e. by taking the ratio of sick($t_{peak}$)/sick($t_{action}$+6).

Calculations were made for 20 Chinese provinces with more than 150 reported cases, and also for China excluding Hubei, by far the worst affected province. The built-in function "SSlogis" in R [24] was used to fit logistic growth curves.

## Results

The cumulative number of cases (confirmed by testing or based on clinical symptoms) was described very well by a logistic growth pattern with $R^2$ greater than 0.99 for all provinces, except Shandong ($R^2$>0.98, S1 Table). Three provinces enacted suppression measures on 23 January, five implemented measures on 24 January, and the remaining 12 provinces started measures on 25 January. The time scale for the increase was $c$ = 3.91 d. for China (excluding Hubei, range 3.13–6.39) and 4.13 d. for Hubei (S1 Table), indicating doubling times of $c * \ln(2)$ = 2.7 d. (range 2.2–4.4 d) and 2.9 d. for China (excluding Hubei) and for Hubei, respectively, during the early epidemic. Some lack of fit during the early phase of the epidemic (before measures) suggests the actual doubling times may be even shorter than these estimates (S1 Table).

The number of reported active sick cases (total infected minus recovered minus deceased) in Hubei peaked 25 days after suppression measures were implemented, which in the model also indicates the peak of number of deaths on the same day, based on the assumption of a fixed daily death rate in the model (delay was not taken into account). Outside Hubei, the peak number of reported sick cases (and peak of daily number dying) was on average reached 18 days after the start of suppression (Table 1). The rate of daily increase in reported cases peaked 17 days after the start of suppression measures in Hubei, and on average 10 days after the start of measures in the other provinces (range: 8 to 15 days). When assuming a reporting delay of 6 days [9], the actual peak in the rate of increase occurred at 11 days after the implementation of measures in Hubei and at 4 days after the implementation of measures in the other provinces. The actual peak in the number of sick cases peaked at 19 days after the start of measures in Hubei and at 12 days after the start of measures in the other provinces.

The relative rate of increase in the number of cases at the time of the peak rate was rather consistent among provinces, with an average of 0.11 cases/case/day for Hubei and 0.12 cases/case/day for the rest of China (range: 0.08 to 0.15). Between suppression implementation and the peak number of reported cases, the number of active cases in Hubei increased by a factor 39, while in other provinces it increased by a factor of 9.5 with considerable variation between provinces (range: 6.2 in Hainan and Chongqing to 20.4 in Heilongjiang). If a 6-day reporting

Table 1. Total confirmed infected cases (up to 10th March), emergency action start date and characteristics of logistic growth curves of epidemic progress in mainland China (excluding Hubei) and other 20 provinces with at least 150 cases of SARS-CoV2. Ratio refers to the sick cases at peak to the total number of infected cases at action date.

| Region | Total confirmed infected cases | Emergency action start date | Peak date of active sick cases (or peak daily death) | Daily increase peak date | Daily increase end date | Delay from action until sick peak | Delay from action until maximum rate | Time from action to the outbreak end | Time from maximum rate till sick peak | Max Sick cases | Cases at time of maximum daily increase | Total infected cases at maximum daily increase date | Relative rate At peak increase rate (cases/ case/d) | Ratio no reporting delay | Ratio 6 days reporting delay |
|---|---|---|---|---|---|---|---|---|---|---|---|---|---|---|---|
| China excluding Hubei | 13004 | 24-Jan | 11-Feb | 3-Feb | 6-Mar | 18 | 10 | 42 | 8 | 9423 | 823 | 6692 | 0.12 | 9.5 | 2.6 |
| Hubei | 67773 | 24-Jan | 18-Feb | 10-Feb | 21-Mar | 25 | 17 | 57 | 8 | 49719 | 4086 | 36796 | 0.11 | 39 | 9.6 |
| Guangdong | 1353 | 23-Jan | 10-Feb | 3-Feb | 24-Feb | 18 | 11 | 32 | 7 | 1009 | 95 | 765 | 0.12 | 13.7 | 3.1 |
| Henan | 1272 | 25-Jan | 10-Feb | 3-Feb | 24-Feb | 16 | 9 | 30 | 7 | 947 | 91 | 659 | 0.14 | 10.0 | 2.4 |
| Zhejiang | 1215 | 23-Jan | 9-Feb | 1-Feb | 20-Feb | 17 | 9 | 28 | 8 | 926 | 95 | 648 | 0.15 | 12.4 | 2.5 |
| Hunan | 1018 | 23-Jan | 9-Feb | 2-Feb | 23-Feb | 17 | 10 | 31 | 7 | 718 | 72 | 516 | 0.14 | 12.7 | 2.9 |
| Anhui | 990 | 24-Jan | 11-Feb | 4-Feb | 24-Feb | 18 | 11 | 31 | 7 | 781 | 70 | 542 | 0.13 | 15.4 | 3.5 |
| Jiangxi | 935 | 24-Jan | 11-Feb | 4-Feb | 23-Feb | 18 | 11 | 30 | 7 | 731 | 70 | 529 | 0.13 | 17.3 | 3.5 |
| Shandong | 758 | 24-Jan | 17-Feb | 8-Feb | 10-Mar | 24 | 15 | 46 | 9 | 451 | 30 | 399 | 0.08 | 6.3 | 2.8 |
| Jiangsu | 631 | 25-Jan | 11-Feb | 4-Feb | 24-Feb | 17 | 10 | 30 | 7 | 451 | 41 | 337 | 0.12 | 8.9 | 2.5 |
| Chongqing | 576 | 24-Jan | 11-Feb | 2-Feb | 24-Feb | 18 | 9 | 31 | 9 | 416 | 33 | 291 | 0.11 | 6.2 | 2.1 |
| Sichuan | 539 | 24-Jan | 13-Feb | 4-Feb | 26-Feb | 20 | 11 | 33 | 9 | 373 | 28 | 291 | 0.10 | 6.7 | 2.4 |
| Heilongjiang | 482 | 25-Jan | 14-Feb | 6-Feb | 25-Feb | 20 | 12 | 31 | 8 | 374 | 33 | 247 | 0.13 | 20.4 | 4.6 |
| Beijing | 435 | 24-Jan | 11-Feb | 3-Feb | 23-Feb | 18 | 10 | 30 | 8 | 299 | 24 | 223 | 0.11 | 6.8 | 2.3 |
| Shanghai | 344 | 24-Jan | 9-Feb | 2-Feb | 19-Feb | 16 | 9 | 26 | 7 | 261 | 23 | 190 | 0.12 | 7.7 | 2.1 |
| Hebei | 318 | 24-Jan | 12-Feb | 5-Feb | 25-Feb | 19 | 12 | 32 | 7 | 207 | 18 | 161 | 0.11 | 10.3 | 3.1 |
| Fujian | 296 | 24-Jan | 10-Feb | 1-Feb | 18-Feb | 17 | 8 | 25 | 9 | 238 | 21 | 149 | 0.14 | 8.2 | 2.2 |
| Guangxi | 252 | 24-Jan | 12-Feb | 3-Feb | 21-Feb | 19 | 10 | 28 | 9 | 195 | 14 | 137 | 0.10 | 7.1 | 2.4 |
| Shaanxi | 245 | 25-Jan | 11-Feb | 2-Feb | 19-Feb | 17 | 8 | 25 | 9 | 197 | 16 | 126 | 0.12 | 6.8 | 2.1 |
| Yunnan | 174 | 24-Jan | 10-Feb | 1-Feb | 16-Feb | 17 | 8 | 23 | 9 | 145 | 11 | 91 | 0.13 | 7.1 | 2.1 |
| Hainan | 168 | 25-Jan | 11-Feb | 4-Feb | 20-Feb | 17 | 10 | 26 | 7 | 120 | 10 | 94 | 0.10 | 6.2 | 2.1 |

delay is accounted for in the estimate of the factor increase from the start of measures to the peak, then this multiplication factor is diminished to a value of 9.5 for Hubei and an average of 2.6 for the other provinces. The daily death rate of the active sick people was 0.34 percent per day for Hubei, much higher than in other provinces (on average 0.05 percent per day, ranging between 0 in Jiangsu Province (see also Sun, Qiu [25] and 0.18 percent per day in Hainan Province, S1 Table), which indicates that on the day of peak sick in Hubei (about 50,000 reported cases), 170 individuals died. Cumulative to 10 March, the modelled logistic curve showed that for China (excluding Hubei) 0.85% of the reported cases died which is close to the actual value of 0.86%. The death rate was overestimated in Hubei as 6.7%, compared to the actual 4.5%, due to the actual daily death rate declining during the later stages of the epidemic (S1 Fig).

Modelled logistic curves show that the total number of infected cases had plateaued by 21 March for Hubei and by 6 March for the rest of mainland China, i.e. 57 and 42 days (range: 23 to 46 days) after the start of suppression measures (Fig 1, Table 1).

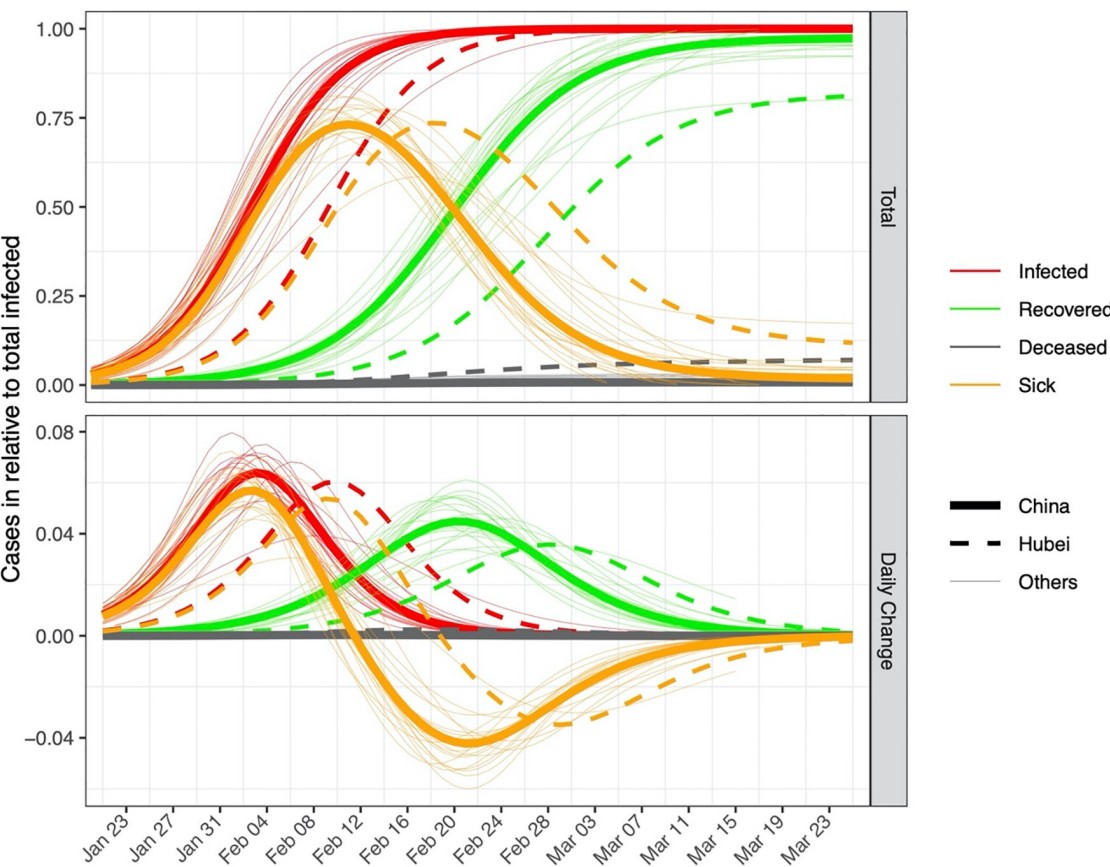

**Fig 1. Fitted epidemic curves based on the observed data of SARS-CoV2 in China excluding Hubei (thick solid lines), Hubei (dashed lines), and 19 other provinces (thin solid lines).** The y-axis of the top panel shows the number of cases relative to the maximum cumulative infected for each region (the value *a*, see Methods). The bottom panel shows the daily change on the same relative scale. Red, green and grey colors indicate confirmed, recovered and deceased cases. Orange color indicates the number of "active" sick cases (relative to total infected, top panel), i.e. infected and not yet recovered or deceased, and the daily changes (bottom panel), with negative values in the lower panel indicating that the number of active cases is decreasing. Fitted and observed values for the true number can be found in S2 and S3 Figs.

## Discussion

Our results show compelling evidence for suppression of SARS-CoV-2 transmission in China, both in Hubei and the other provinces. No new cases were reported within mainland China on 18, 19 and 20 March, 2020, with all new cases on those days from returning overseas travelers, signaling the beginning of the end of the outbreak. However isolated cases still occur, e.g. there was one new case on 24 March in Hubei. Thus, successful suppression need to be followed by measures that will prevent a new outbreak [11].

Here we used logistic models to study the disease dynamic parameters. The analysis shown in our results was continually updated from 1 February to 3 March while the epidemic was progressing [26]. Based on the data up till 16 February 2020, a peak in sick incidence was identified for 12 February and the near ending of new confirmed cases was 23 February, which was proven correct [27]. Thus, logistic models may be used to determine early when suppression measures are expected to result in decreased rate of epidemic growth and a decline in number of sick cases. However, any model shows lack of fit [18]. The logistic model did not capture that the rate of increase in the early epidemic is faster than the rate of decline during the tapering out of the epidemic (S2 Fig). Thus, the logistic underestimates both the early relative growth rate and the increase of the number of sick people from the start of measures till the peak for the Chinese data (S2 Fig). Uncertainties in predictions also result from unknown reporting delay [9]. Improvements may be possible by defining better tailored models, considering a delay between development of symptoms and mortality (as in classical epidemic models), and especially, by collecting better data, e.g. more (random) testing.

The results show that suppression can lead to (almost) complete removal of active virus infected cases from the population, although given that not all active cases have recovered, the outbreak is not completely over (as of 8 April 2020). The virus could still be present in asymptomatic individuals or it could be re-introduced from unknown reservoirs. Re-entry of the virus from countries outside China needs to be prevented. Because the vast majority of the population is not immune to SARS-CoV2, the virus can rapidly re-establish. Therefore, suppression needs to be followed up by containment, a strategy based on strict surveillance, testing of all individuals with symptoms, and followed by isolation of all infected individuals and their recent contacts [11]. Currently, as of 8 April 2020, quarantine restrictions are gradually being lifted in China, including Hubei Province, after no new cases have been detected for several weeks, allowing people to return to work and businesses to start up again. Moreover, in China, schools are preparing to reopen in April and normal social activities are slowly resuming [28]. These findings indicate that the implemented measures have been effective for controlling SARS-CoV2 transmission in China [23, 29–31].

Many individuals infected with SARS-CoV2 show minimal or no symptoms. Due in part to asymptomatic carriers, many infected individuals remain untested and unreported. Before 31 March, asymptomatic cases were not included in in the outbreak's daily report in China [32]. It is estimated that unreported cases were responsible for 77% of the reproduction number of the disease before the start of measures in China and 16–21% thereafter [9, 10]. Moreover, about half of infected individuals that develop symptoms do not show symptoms until 5 days after infection, and some not up to 14 days, and maybe even up to 30 days [10, 33]. Cases with a long incubation period, if they exist, could contribute to re-emergence of COVID-19 after restrictions are lifted. Nonetheless, as the experience in China and several other countries/ regions (e.g. South Korea, Taiwan, Hongkong, Japan and Singapore) has shown, a containment strategy can prevent the virus from uncontrolled spread if it re-emerges. Given the worldwide pandemic spread of SARS-CoV2, it seems increasingly unavoidable that worldwide

containment will depend on a vaccine. Until a vaccine is ready and accessible, the world population must confront the pandemic by combining suppression and containment in a practical way that minimizes the human and economic costs. As noted by Wu and McGoogan (8), "It is not only individual rights that need to be considered. The rights of those who are not infected, but at risk of infection, must be considered as well."

Important lessons from the outbreak and its control in China are in our opinion:

1. Suppression of SARS-CoV-2 is possible even after widespread community transmission.

2. Suppression can be achieved in one to two months if stringent measures are implemented and maintained.

3. If implementation of stringent suppression measures is delayed, as was the case in Hubei, the peak outbreak time is later, the increase in the number of sick people is greater, the number of people dying is higher, and the necessary period of suppression is longer.

4. China provides compelling evidence that suppression of SARS-CoV-2 transmission can be achieved within 60 days, even following widespread community transmission. It is the opinion of the authors that addressing the widespread SARS-CoV-2 transmission in other countries with unproven mitigation strategies may subject a large part of their populations unnecessarily to the adverse health risks associated with COVID-19 [7, 34].

## Supporting information

**S1 Appendix. Formula inference.**
(PDF)

**S1 Table. Estimated model parameters per province and for China (excluding Hubei) and other 20 provinces with at least 150 cases of SARS-CoV2.**
(PDF)

**S1 Fig. Daily death rate for Hubei province (a) and China excluding Hubei (b).**
(PDF)

**S2 Fig. Observed and fitted epidemics of SARS-CoV2 in mainland China excluding Hubei province for normal- (a) and log-scale (b) y-axis.** Dots are observed cases and lines are model fits. Top panel refers total cases and bottom panel refers to daily changes. For the normal scale (A), the left y-axis is for infected, recovered and sick cases and the right y-axis for deceased cases. Red colour indicates the total number of infected cases (confirmed and suspected) (top panel) or the daily rate of increase in the number of cases (bottom panel). Green colour indicate the recovered cases in both panels. Grey colour indicates cumulative deaths in the upper graph and daily death cases in the lower panel. Orange colour indicates the number of "active" cases (top panel), i.e. infected and not yet recovered or deceased, and the daily change in the number of active cases (bottom panel). Negative values (in normal scale) in the lower panel mean that the number of active cases is decreasing.
(PDF)

**S3 Fig. Epidemics of SARS-CoV2 in 20 Chinese provinces with a minimum of 150 cases for normal- (a) and log-scale (b) y-axis.** Dots are observed cases and lines are model fits. Red, green and grey colors indicate confirmed, recovered and deceased cases. For the normal scale (a), the left y-axis is for infected, recovered and sick cases and the right y-axis for deceased cases. Orange color indicates the number of "active" sick cases (relative to total infected, top panel), i.e. infected and not yet recovered or deceased, and the daily changes (bottom panel),

with negative values (in normal scale) in the lower panel indicating that the number of active cases is decreasing. The upper panel for each province refers to the total number of cases while the lower panel refers to the daily change in the number of cases.
(PDF)

## Acknowledgments

We thank Jiajun Liu, Elmer Villanueva, Kevin Schneider, David Kottelenberg, Wytse van der Werf, Bouke van der Werf, Marco Pautasso, and Hans Heesterbeek for comments and suggestions.

## Author Contributions

**Conceptualization:** Yi Zou, Peng Zhao, Lei Han, Xiaoxiang Wang, Johannes Knops, Wopke van der Werf.

**Formal analysis:** Yi Zou.

**Methodology:** Lia Hemerik, Wopke van der Werf.

**Software:** Yi Zou, Peng Zhao.

**Writing – original draft:** Yi Zou, Stephen Pan, Johannes Knops, Wopke van der Werf.

**Writing – review & editing:** Yi Zou, Stephen Pan, Xiaoxiang Wang, Lia Hemerik, Johannes Knops, Wopke van der Werf.

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
