## [Decision Letter · Decision Letter 0]

27 May 2020

PONE-D-20-10245

Outbreak analysis with a logistic growth model shows COVID-19 suppression dynamics in China

PLOS ONE

Dear Dr. Zou,

Thank you for submitting your manuscript to PLOS ONE. After careful consideration, we feel that it has merit but does not fully meet PLOS ONE’s publication criteria as it currently stands. Therefore, we invite you to submit a revised version of the manuscript that addresses the points raised during the review process.

We look forward to receiving your revised manuscript.

Kind regards,

Abdallah M. Samy, PhD

Academic Editor

PLOS ONE

**Journal Requirements:**

2. In ethics statement in the manuscript and in the online submission form, please provide additional information about the database used in your retrospective study. Specifically, please ensure that you have discussed whether all data were fully anonymized before you accessed them and/or whether the IRB or ethics committee waived the requirement for informed consent. If patients provided informed written consent to have their data used in research, please include this information.

**Reviewers' comments:**

Reviewer's Responses to Questions

**Comments to the Author**

1. Is the manuscript technically sound, and do the data support the conclusions?

Reviewer #1: Yes

2. Has the statistical analysis been performed appropriately and rigorously? 

Reviewer #1: Yes

3. Have the authors made all data underlying the findings in their manuscript fully available?

Reviewer #1: Yes

4. Is the manuscript presented in an intelligible fashion and written in standard English?

Reviewer #1: Yes

5. Review Comments to the Author

Reviewer #1: Introduction

Line 64-65

“People needs hospitalization or intensive care=20%”

Infected people requiring intensive care are around 5%, please add this information (with reference) to avoid confusion with the 20% you reported for confirmed infections requiring hospitalization.

In addition, this information (confirmed infections requiring hospitalization = 20%) varies according to different studies, it would be preferable to add other percentages from other studies with references (optional).

Methods

Line 131-132

You said “you have calculated the peak date of active sick people, which is also the date of peak number of daily death”,

1. Is this a fact?

2. Is the date peak of death match always the peak date of sick (infected) people?

3. What about the interval of time between sicknesses in death?

Please explain this point.

Line 132

“2) The number of sick cases during the peak”, is it the cases number only on the peak duration or the cumulative cases number of all the previous period?

Please to clarify that.

Results

Line 164-165

You said “which in the model also indicates the peak of number of deaths on the same day”

This is related to the remark we made to you above about the interval between illness and death.

6. PLOS authors have the option to publish the peer review history of their article (what does this mean?). If published, this will include your full peer review and any attached files.

Reviewer #1: Yes: Mohamed HAMIDOUCHE

---

## [Author Response · Author response to Decision Letter 0]

4 Jun 2020

Response to Editor

1. Before we can proceed with your paper, please provide more information about the reports you used from the National Health Commission of China.

In the Methods section of your manuscript:

a) Please clarify whether or not the data was fully anonymized before you accessed them and/or whether the IRB or ethics committee waived the requirement for informed consent.

b) If patients provided informed written consent to have their data used in research, please include this information.

R: This does not apply to our study. Our data does not contain data related patient. In method we have now modified “Data were obtained from the open-source analytics tools, the R package “nCov2019” [19, 20], which retrieves data from the daily epidemic report in the National Health Commission of China [21]” (L127-129). We also updated the reference of 21. 

In the Data Report, we changed to “Data used in this study are publicly accessible from the R package nCov2019 (https://github.com/GuangchuangYu/nCov2019).” We hope it is clear now. 

Response to Reviewer

Reviewer #1: Introduction

Line 64-65

“People needs hospitalization or intensive care=20%”

Infected people requiring intensive care are around 5%, please add this information (with reference) to avoid confusion with the 20% you reported for confirmed infections requiring hospitalization.

In addition, this information (confirmed infections requiring hospitalization = 20%) varies according to different studies, it would be preferable to add other percentages from other studies with references (optional).

R: We have rewritten this part of the paper to reflect more clearly that the high proportion of 20% is the proportion of reported cases requiring hospitalization, but the reporting fraction is quite variable over countries and regions according to testing capacity, definition of a positive case on symptoms or only on testing results, and so on. We also state that the proportion of infected people requiring hospitalization is at least 3% according to the available knowledge, but is still uncertain because the reporting fraction is poorly identified. The given lower bound accords well with the reviewer’s assessment, with which we agree. References are included to reflect the uncertainty in reporting.

Methods

Line 131-132

You said “you have calculated the peak date of active sick people, which is also the date of peak number of daily death”,

1. Is this a fact?

2. Is the date peak of death match always the peak date of sick (infected) people?

3. What about the interval of time between sicknesses in death?

Please explain this point.

R: We clarified as “active sick people, which is also the date of peak number of daily death as we assumed a fixed daily death rate in the model” (L163-165 in the tracked change version). In addition, we added a new Supplementary information (S1 Fig. 1) to indicate the daily death rate which was relative constant. However, we also mention in L225-227, “The death rate was overestimated in Hubei as 6.7%, compared to the actual 4.5%, due to the actual daily death rate declining during the later stages of the epidemic (S1 Fig. 1)”

We added in the methods a cautionary remark “The model does not account for the delay between the moment of reporting disease and the moment of reporting death.” And we wrote in the discussion “The number of reported active sick cases (total infected minus recovered minus deceased) in Hubei peaked 25 days after suppression measures were implemented, which in the model also indicates the peak of number of deaths on the same day, based on the assumption of a fixed daily death rate in the model (delay was not taken into account)”. Thus, we clearly acknowledge that the assumed proportionality between the number of active cases and the number dying is a simplifying assumption.

Line 132

“2) The number of sick cases during the peak”, is it the cases number only on the peak duration or the cumulative cases number of all the previous period?

Please to clarify that.

R: We changed as number of active cases sick cases during the peak

Results

Line 164-165

You said “which in the model also indicates the peak of number of deaths on the same day”

This is related to the remark we made to you above about the interval between illness and death.

R: We added “…based on the assumption of a fixed daily death rate in the model” (L197-198)

---

## [Editor Report · Decision Letter 1]

12 Jun 2020

Outbreak analysis with a logistic growth model shows COVID-19 suppression dynamics in China

PONE-D-20-10245R1

Dear Dr. Zou,

We’re pleased to inform you that your manuscript has been judged scientifically suitable for publication and will be formally accepted for publication once it meets all outstanding technical requirements.

Kind regards,

Abdallah M. Samy, PhD

Academic Editor

PLOS ONE

---

## [Editor Report · Acceptance letter]

18 Jun 2020

PONE-D-20-10245R1 

Outbreak analysis with a logistic growth model shows COVID-19 suppression dynamics in China 

Dear Dr. Zou:

I'm pleased to inform you that your manuscript has been deemed suitable for publication in PLOS ONE. Congratulations! Your manuscript is now with our production department. 

Kind regards, 

on behalf of

Dr. Abdallah M. Samy 

Academic Editor

PLOS ONE